# Fast-Slow Recurrent Neural Networks

**Asier Mujika**
Department of Computer Science
ETH Zürich, Switzerland
asierm@ethz.ch

**Florian Meier**
Department of Computer Science
ETH Zürich, Switzerland
meierflo@inf.ethz.ch

**Angelika Steger**
Department of Computer Science
ETH Zürich, Switzerland
steger@inf.ethz.ch

## Abstract

Processing sequential data of variable length is a major challenge in a wide range of applications, such as speech recognition, language modeling, generative image modeling and machine translation. Here, we address this challenge by proposing a novel recurrent neural network (RNN) architecture, the Fast-Slow RNN (FS-RNN). The FS-RNN incorporates the strengths of both multiscale RNNs and deep transition RNNs as it processes sequential data on different timescales and learns complex transition functions from one time step to the next. We evaluate the FS-RNN on two character level language modeling data sets, Penn Treebank and Hutter Prize Wikipedia, where we improve state of the art results to 1.19 and 1.25 bits-per-character (BPC), respectively. In addition, an ensemble of two FS-RNNs achieves 1.20 BPC on Hutter Prize Wikipedia outperforming the best known compression algorithm with respect to the BPC measure. We also present an empirical investigation of the learning and network dynamics of the FS-RNN, which explains the improved performance compared to other RNN architectures. Our approach is general as any kind of RNN cell is a possible building block for the FS-RNN architecture, and thus can be flexibly applied to different tasks.

## 1 Introduction

Processing, modeling and predicting sequential data of variable length is a major challenge in the field of machine learning. In recent years, recurrent neural networks (RNNs) [34, 32, 39, 41] have been the most popular tool to approach this challenge. RNNs have been successfully applied to improve state of the art results in complex tasks like language modeling and speech recognition. A popular variation of RNNs are long short-term memories (LSTMs) [18], which have been proposed to address the vanishing gradient problem [16, 5, 17]. LSTMs maintain constant error flow and thus are more suitable to learn long-term dependencies compared to standard RNNs.

Our work contributes to the ongoing debate on how to interconnect several RNN cells with the goals of promoting the learning of long-term dependencies, favoring efficient hierarchical representations of information, exploiting the computational advantages of deep over shallow networks and increasing computational efficiency of training and testing. In deep RNN architectures, RNNs or LSTMs are stacked layer-wise on top of each other [9, 20, 11]. The additional layers enable the network to learn complex input to output relations and encourage a efficient hierarchical representation of information. In these architectures, the hidden states of all the hierarchical layers are updated once per time step (by one time step we refer to the time between two consecutive input elements). In multiscale RNN architectures [35, 9, 25, 6], the operation on different timescales is enforced

by updating the higher layers less frequently, which further encourages an efficient hierarchical representation of information. Updating higher layers in fewer time steps leads to computationally efficient implementations and gives rise to short gradient paths that favor the learning of long-term dependencies. In deep transition RNN architectures, intermediate sequentially connected layers are interposed between two consecutive hidden states in order to increase the depth of the transition function from one time step to the next, as for example in deep transition networks [31] or Recurrent Highway Networks (RHN) [43]. The intermediate layers enable the network to learn complex non-linear transition functions. Thus, the model exploits the fact that deep models can represent some functions exponentially more efficiently than shallow models [4]. We interpret these networks as several RNN cells that update a single hidden state sequentially. Observe that any RNN cell can be used to build a deep transition RNN by connecting several of these cells sequentially.

Here, we propose the Fast-Slow RNN (FS-RNN) architecture, a novel way of interconnecting RNN cells, that combines advantages of multiscale RNNs and deep transition RNNs. The architecture consists of $k$ sequentially connected RNN cells in the lower hierarchical layer and one RNN cell in the higher hierarchical layer, see Figure 1 and Section 3. Therefore, the hidden state of the lower layer is updated $k$ times per time step, whereas the hidden state of the higher layer is updated only once per time step. We evaluate the FS-RNN on two standard character level language modeling data sets, namely Penn Treebank and Hutter Prize Wikipedia. Additionally, following [31], we present an empirical analysis that reveals advantages of the FS-RNN architecture over other RNN architectures.

The main contributions of this paper are:

- We propose the FS-RNN as a novel RNN architecture.
- We improve state of the art results on the Penn Treebank and Hutter Prize Wikipedia data sets.
- We surpass the BPC performance of the best known text compression algorithm evaluated on Hutter Prize Wikipedia by using an ensemble of two FS-RNNs.
- We show empirically that the FS-RNN incorporates strengths of both multiscale RNNs and deep transition RNNs, as it stores long-term dependencies efficiently and it adapts quickly to unexpected input.
- We provide our code in the following URL https://github.com/amujika/Fast-Slow-LSTM.

## 2 Related work

In the following, we review the work that relates to our approach in more detail. First, we focus on deep transition RNNs and multiscale RNNs since these two architectures are the main sources of inspiration for the FS-RNN architecture. Then, we discuss how our approach differs from these two architectures. Finally, we review other approaches that address the issue of learning long-term dependencies when processing sequential data.

Pascanu et al. [31] investigated how a RNN can be converted into a deep RNN. In standard RNNs, the transition function from one hidden state to the next is shallow, that is, the function can be written as one linear transformation concatenated with a point wise non-linearity. The authors added intermediate layers to increase the depth of the transition function, and they found empirically that such deeper architectures boost performance. Since deeper architectures are more difficult to train, they equip the network with skip connections, which give rise to shorter gradient paths (DT(S)-RNN, see [31]). Following a similar line of research, Zilly et al. [43] further increased the transition depth between two consecutive hidden states. They used highway layers [38] to address the issue of training deep architectures. The resulting RHN [43] achieved state of the art results on the Penn Treebank and Hutter Prize Wikipedia data sets. Furthermore, a vague similarity to deep transition networks can be seen in adaptive computation [12], where an LSTM cell learns how many times it should update its state after receiving the input to produce the next output.

Multiscale RNNs are obtained by stacking multiple RNNs with decreasing order of update frequencies on top of each other. Early attempts proposed such architectures for sequential data compression [35], where the higher layer is only updated in case of prediction errors of the lower layer, and for sequence classification [9], where the higher layers are updated with a fixed smaller frequency. More recently, Koutnik et al. [25] proposed the Clockwork RNN, in which the hidden units are divided into

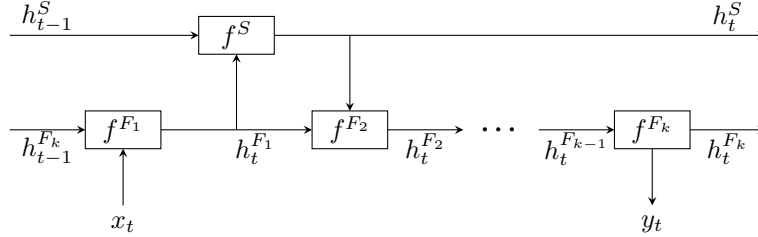

Figure 1: Diagram of a Fast-Slow RNN with $k$ Fast cells. Observe that only the second Fast cell receives the input from the Slow cell.

several modules, of which the $i$-th module is only updated every $2^i$-th time-step. General advantages of this multiscale RNN architecture are improved computational efficiency, efficient propagation of long-term dependencies and flexibility in allocating resources (units) to the hierarchical layers. Multiscale RNNs have been applied for speech recognition in [3], where the slower operating RNN pools information over time and the timescales are fixed hyperparameters as in Clockwork RNNs. In [36], multiscale RNNs are applied to make context-aware query suggestions. In this case, explicit hierarchical boundary information is provided. Chung et al. [6] presented a hierarchical multiscale RNN (HM-RNN) that discovers the latent hierarchical structure of the sequence without explicitly given boundary information. If a parametrized boundary detector indicates the end of a segment, then a summarized representation of the segment is fed to the upper layer and the state of the lower layer is reset [6].

Our FS-RNN architectures borrows elements from both deep transition RNNs and multiscale RNNs. The lower hierarchical layer is a deep transition RNN, that updates the hidden state several times per time step, whereas the higher hierarchical layer updates the hidden state only once per time step.

Many approaches aim at solving the problem of learning long-term dependencies in sequential data. A very popular one is to use external memory cells that can be accessed and modified by the network, see Neural Turing Machines [13], Memory Networks [40] and Differentiable Neural Computer [14]. Other approaches focus on different optimization techniques rather than network architectures. One attempt is Hessian Free optimization [29], a second order training method that achieved good results on RNNs. The use of different optimization techniques can improve learning in a wide range of RNN architectures and therefore, the FS-RNN may also benefit from it.

## 3   Fast-Slow RNN

We propose the FS-RNN architecture, see Figure 1. It consists of $k$ sequentially connected RNN cells $F_1, \ldots, F_k$ on the lower hierarchical layer and one RNN cell $S$ on the higher hierarchical layer. We call $F_1, \ldots, F_k$ the *Fast* cells, $S$ the *Slow* cell and the corresponding hierarchical layers the *Fast* and *Slow* layer, respectively. $S$ receives input from $F_1$ and feeds its state to $F_2$. $F_1$ receives the sequential input data $x_t$, and $F_k$ outputs the predicted probability distribution $y_t$ of the next element of the sequence.

Intuitively, the Fast cells are able to learn complex transition functions from one time step to the next one. The Slow cell gives rise to shorter gradient paths between sequential inputs that are distant in time, and thus, it facilitates the learning of long-term dependencies. Therefore, the FS-RNN architecture incorporates advantages of deep transition RNNs and of multiscale RNNs, see Section 2.

Since any kind of RNN cell can be used as building block for the FS-RNN architecture, we state the formal update rules of the FS-RNN for arbitrary RNN cells. We define a RNN cell $Q$ to be a differentiable function $f^Q(h, x)$ that maps a hidden state $h$ and an additional input $x$ to a new hidden state. Note that $x$ can be input data or input from a cell in a higher or lower hierarchical layer. If a cell does not receive an additional input, then we will omit $x$. The following equations define the FS-RNN architecture for arbitrary RNN cells $F_1, \ldots, F_k$ and $S$.

$$h_t^{F_1} = f^{F_1}(h_{t-1}^{F_k}, x_t)$$
$$h_t^S = f^S(h_{t-1}^S, h_t^{F_1})$$
$$h_t^{F_2} = f^{F_2}(h_t^{F_1}, h_t^S)$$
$$h_t^{F_i} = f^{F_i}(h_t^{F_{i-1}}) \quad \text{for } 3 \leq i \leq k$$

The output $y_t$ is computed as an affine transformation of $h_t^{F_k}$. It is possible to extend the FS-RNN architecture in order to further facilitate the learning of long-term dependencies by adding hierarchical layers, each of which operates on a slower timescale than the ones below, resembling clockwork RNNs [25]. However, for the tasks considered in Section 4, we observed that this led to overfitting the training data even when applying regularization techniques and reduced the performance at test time. Therefore, we will not further investigate this extension of the model in this paper, even though it might be beneficial for other tasks or larger data sets.

In the experiments in Section 4, we use LSTM cells as building blocks for the FS-RNN architecture. For completeness, we state the update function $f^Q$ for an LSTM $Q$. The state of an LSTM is a pair $(h_t, c_t)$, consisting of the hidden state and the cell state. The function $f^Q$ maps the previous state and input $(h_{t-1}, c_{t-1}, x_t)$ to the next state $(h_t, c_t)$ according to

$$\begin{pmatrix} f_t \\ i_t \\ o_t \\ g_t \end{pmatrix} = W_h^Q h_{t-1} + W_x^Q x_t + b^Q$$
$$c_t = \sigma(f_t) \odot c_{t-1} + \sigma(i_t) \odot \tanh(g_t)$$
$$h_t = \sigma(o_t) \odot \tanh(c_t),$$

where $f_t$, $i_t$ and $o_t$ are commonly referred to as forget, input and output gates, and $g_t$ are the new candidate cell states. Moreover, $W_h^Q$, $W_x^Q$ and $b^Q$ are the learnable parameters, $\sigma$ denotes the sigmoid function, and $\odot$ denotes the element-wise multiplication.

## 4   Experiments

For the experiments, we consider the Fast-Slow LSTM (FS-LSTM) that is a FS-RNN, where each RNN cell is a LSTM cell. The FS-LSTM is evaluated on two character level language modeling data sets, namely Penn Treebank and Hutter Prize Wikipedia, which will be referred to as $enwik8$ in this section. The task consists of predicting the probability distribution of the next character given all the previous ones. In Section 4.1, we compare the performance of the FS-LSTM with other approaches. In Section 4.2, we empirically compare the network dynamics of different RNN architectures and show the FS-LSTM combines the benefits of both, deep transition RNNs and multiscale RNNs.

### 4.1   Performance on Penn Treebank and Hutter Prize Wikipedia

The FS-LSTM achieves 1.19 BPC and 1.25 BPC on the Penn Treebank and $enwik8$ data sets, respectively. These results are compared to other approaches in Table 1 and Table 2 (the baseline LSTM results without citations are taken from [44] for Penn Treebank and from [15] for $enwik8$). For the Penn Treebank, the FS-LSTM outperforms all previous approaches with significantly less parameters than the previous top approaches. We did not observe any improvement when increasing the model size, probably due to overfitting. In the $enwik8$ data set, the FS-LSTM surpasses all other neural approaches. Following [13], we compare the results with text compression algorithms using the BPC measure. An ensemble of two FS-LSTM models (1.20 BPC) outperforms cmix (1.23 BPC) [24], the current best text compression algorithm on $enwik8$ [27]. However, a fair comparison is difficult. Compression algorithms are usually evaluated by the final size of the compressed data set including the decompressor size. For character prediction models, the network size is usually not taken into account and the performance is measured on the test set. We remark that as the FS-LSTM is evaluated on the test set, it should achieve similar performance on any part of the English Wikipedia.

Table 1: BPC on Penn Treebank

| Model | BPC | Param Count |
|---|---|---|
| Zoneout LSTM [2] | 1.27 | - |
| 2-Layers LSTM | 1.243 | 6.6M |
| HM-LSTM [6] | 1.24 | - |
| HyperLSTM - small [15] | 1.233 | 5.1M |
| HyperLSTM [15] | 1.219 | 14.4M |
| NASCell - small [44] | 1.228 | 6.6M |
| NASCell [44] | 1.214 | 16.3M |
| FS-LSTM-2 (ours) | 1.190 | 7.2M |
| FS-LSTM-4 (ours) | 1.193 | 6.5M |

The FS-LSTM-2 and FS-LSTM-4 model consist of two and four cells in the Fast layer, respectively. The FS-LSTM-4 model outperforms the FS-LSTM-2 model, but its processing time for one time step is 25% higher than the one of the FS-LSTM-2. Adding more cells to the Fast layer could further improve the performance as observed for RHN [43], but would increase the processing time, because the cell states are computed sequentially. Therefore, we did not further increase the number of Fast cells.

The model is trained to minimize the cross-entropy loss between the predictions and the training data. Formally, the loss function is defined as $L = -\frac{1}{n} \sum_{i=1}^{n} \log p_\theta(x_i|x_1, \ldots, x_{i-1})$, where $p_\theta(x_i|x_1, \ldots, x_{i-1})$ is the probability that a model with parameters $\theta$ assigns to the next character $x_i$ given all the previous ones. The model is evaluated by the BPC measure, which uses the binary logarithm instead of the natural logarithm in the loss function. All the hyperparameters used for the experiments are summarized in Table 3. We regularize the FS-LSTM with dropout [37]. In each time step, a different dropout mask is applied for the non-recurrent connections [42], and Zoneout [2] is applied for the recurrent connections. The network is trained with minibatch gradient descent using the Adam optimizer [23]. If the gradients have norm larger than 1 they are normalized to 1. Truncated backpropagation through time (TBPTT) [34, 10] is used to approximate the gradients, and the final hidden state is passed to the next sequence. The learning rate is divided by a factor 10 for the last 20 epochs in the Penn Treebank experiments, and it is divided by a factor 10 whenever the validation error does not improve in two consecutive epochs in the $enwik8$ experiments. The forget bias of every LSTM cell is initialized to 1, and all weight matrices are initialized to orthogonal matrices. Layer normalization [1] is applied to the cell and to each gate separately. The network with the smallest validation error is evaluated on the test set. The two data sets that we use for evaluation are:

**Penn Treebank [28]**   The dataset is a collection of Wall Street Journal articles written in English. It only contains 10000 different words, all written in lower-case, and rare words are replaced with "$< unk >$". Following [30], we split the data set into train, validation and test sets consisting of 5.1M, 400K and 450K characters, respectively.

**Hutter Prize Wikipedia [19]**   This dataset is also known as $enwik8$ and it consists of "raw" Wikipedia data, that is, English articles, tables, XML data, hyperlinks and special characters. The data set contains 100M characters with 205 unique tokens. Following [7], we split the data set into train, validation and test sets consisting of 90M, 5M and 5M characters, respectively.

## 4.2   Comparison of network dynamics of different architectures

We compare the FS-LSTM architecture with the stacked-LSTM and the sequential-LSTM architectures, depicted in Figure 2, by investigating the network dynamics. In order to conduct a fair comparison we chose the number of parameters to roughly be the same for all three models. The FS-LSTM consists of one Slow and four Fast LSTM cells of 450 units each. The stacked-LSTM consists of five LSTM cells stacked on top of each other consisting of 375 units each, which will be

Table 2: BPC on $enwik8$

| Model | BPC | Param Count |
|---|---|---|
| LSTM, 2000 units | 1.461 | 18M |
| Layer Norm LSTM, 1800 units | 1.402 | 14M |
| HyperLSTM [15] | 1.340 | 27M |
| HM-LSTM [6] | 1.32 | 35M |
| Surprisal-driven Zoneout [33] | 1.31 | 64M |
| ByteNet [22] | 1.31 | - |
| RHN - depth 5 [43] | 1.31 | 23M |
| RHN - depth 10 [43] | 1.30 | 21M |
| Large RHN - depth 10 [43] | 1.27 | 46M |
| FS-LSTM-2 (ours) | 1.290 | 27M |
| FS-LSTM-4 (ours) | 1.277 | 27M |
| Large FS-LSTM-4 (ours) | 1.245 | 47M |
| $2 \times$ Large FS-LSTM-4 (ours) | 1.198 | $2 \times$ 47M |
| cmix v13 [24] | 1.225 | - |

Table 3: Hyperparameters for the character-level language model experiments.

| | Penn Treebank | | $enwik8$ | | |
|---|---|---|---|---|---|
| | FS-LSTM-2 | FS-LSTM-4 | FS-LSTM-2 | FS-LSTM-4 | Large FS-LSTM-4 |
| Non-recurrent dropout | 0.35 | 0.35 | 0.2 | 0.2 | 0.25 |
| Cell zoneout | 0.5 | 0.5 | 0.3 | 0.3 | 0.3 |
| Hidden zoneout | 0.1 | 0.1 | 0.05 | 0.05 | 0.05 |
| Fast cell size | 700 | 500 | 900 | 730 | 1200 |
| Slow cell size | 400 | 400 | 1500 | 1500 | 1500 |
| TBPTT length | 150 | 150 | 150 | 150 | 100 |
| Minibatch size | 128 | 128 | 128 | 128 | 128 |
| Input embedding size | 128 | 128 | 256 | 256 | 256 |
| Initial Learning rate | 0.002 | 0.002 | 0.001 | 0.001 | 0.001 |
| Epochs | 200 | 200 | 35 | 35 | 50 |

referred to as Stacked-1, ... , Stacked-5, from bottom to top. The sequential-LSTM consists of five sequentially connected LSTM cells of 500 units each. All three models require roughly the same time to process one time step. The models are trained on $enwik8$ for 20 epochs with minibatch gradient descent using the Adam optimizer [23] without any regularization, but layer normalization [1] is applied on the cell states of the LSTMs. The hyperparameters are not optimized for any of the three models. We repeat each experiment 5 times and report the mean and standard deviation.

The experiments suggest that the FS-LSTM architecture favors the learning of long-term dependencies (Figure 3), enforces hidden cell states to change at different rates (Figure 4) and facilitates a quick adaptation to unexpected inputs (Figure 5). Moreover, the FS-LSTM achieves a mean performance of 1.49 BPC with a standard deviation of 0.007 BPC and outperforms the stacked-LSTM (mean 1.60 BPC, standard deviation 0.022 BPC ) and the sequential-LSTM (mean 1.58 BPC, standard deviation 0.008 BPC ).

In Figure 3, we asses the ability to capture long-term dependencies by investigating the effect of the cell state on the loss at later time points, following [2]. We measure the effect of the cell state at time $t - k$ on the loss at time $t$ by the gradient $\|\frac{\partial L_t}{\partial c_{t-k}}\|$. This gradient is the largest for the Slow

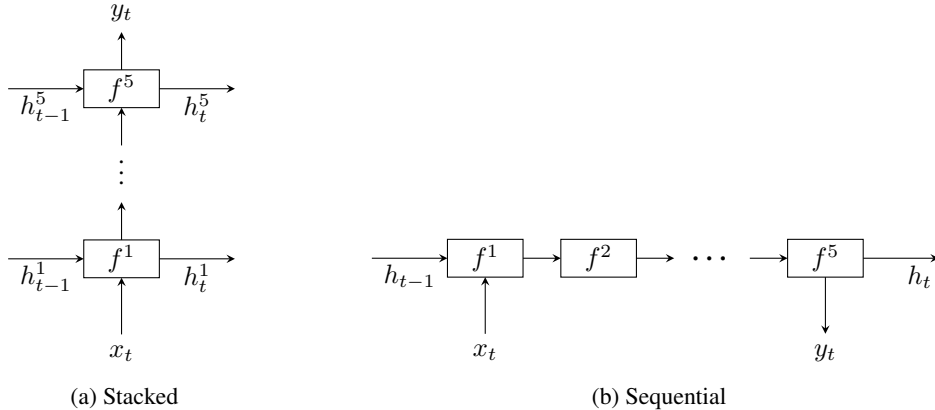

<div align="center">(a) Stacked          (b) Sequential</div>

Figure 2: Diagram of (a) stacked-LSTM and (b) sequential-LSTM with 5 cells each.

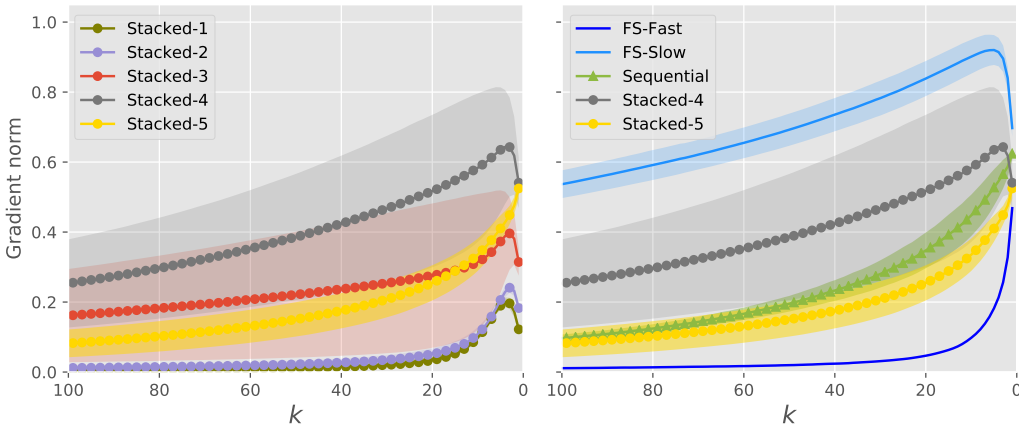

Figure 3: Long-term effect of the cell states on the loss function. The average value of $\left\lVert \frac{\partial L_t}{\partial c_{t-k}} \right\rVert$, which is the effect of the cell state at time $t-k$ on the loss function at time $t$, is plotted against $k$ for the different layers in the three RNN architectures. The shaded area shows the standard deviation. For the sequential-LSTM only the first cell is considered.

LSTM, and it is small and steeply decaying as $k$ increases for the Fast LSTM. Evidently, the Slow cell captures long-term dependencies, whereas the Fast cell only stores short-term information. In the stacked-LSTM, the gradients decrease from the top layer to the bottom layer, which can be explained by the vanishing gradient problem. The small, steeply decaying gradients of the sequential-LSTM indicate that it is less capable to learn long-term dependencies than the other two models.

Figure 4 gives further evidence that the FS-LSTM stores long-term dependencies efficiently in the Slow LSTM cell. It shows that among all the layers of the three RNN architectures, the cell states of the Slow LSTM change the least from one time step to the next. The highest change is observed for the cells of the sequential model followed by the Fast LSTM cells.

In Figure 5, we investigate whether the FS-LSTM quickly adapts to unexpected characters, that is, whether it performs well on the subsequent ones. In text modeling, the initial character of a word has the highest entropy, whereas later characters in a word are usually less ambiguous [10]. Since the first character of a word is the most difficult one to predict, the performance at the following positions should reflect the ability to adapt to unexpected inputs. While the prediction qualities at the first position are rather close for all three models, the FS-LSTM outperforms the stacked-LSTM and sequential-LSTM significantly on subsequent positions. It is possible that new information is incorporated quickly in the Fast layer, because it only stores short-term information, see Figure 3.

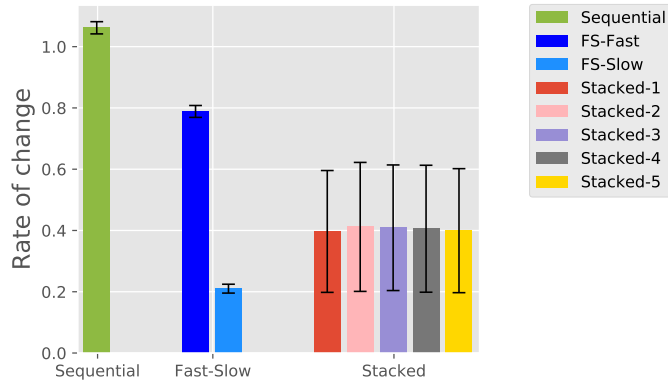

Figure 4: Rate of change of the cell states from one time step to the next. We plot $\frac{1}{n}\sum_{i=1}^{n}(c_{t,i} - c_{t-1,i})^2$ averaged over all time steps, where $c_{t,i}$ is the value of the $i$th unit at time step $t$, for the different layers of the three RNN architectures. The error bars show the standard deviation. For the sequential-LSTM only the first cell is considered.

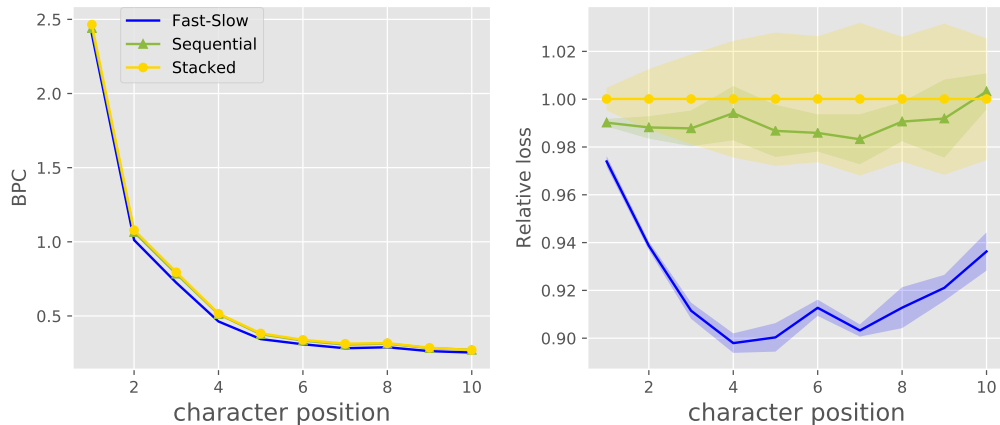

Figure 5: Bits-per-character at each character position. The left panel shows the average bits-per-character at each character positions in the test set. The right panel shows the average relative loss with respect to the stacked-LSTM at each character position. The shaded area shows the standard deviation. For this Figure, a word is considered to be a sequence of lower-case letters of length at least 2 in-between two spaces.

## 5  Conclusion

In this paper, we have proposed the FS-RNN architecture. Up to our knowledge, it is the first architecture that incorporates ideas of both multiscale and deep transition RNNs. The FS-RNN architecture improved state of the art results on character level language modeling evaluated on the Penn Treebank and Hutter Prize Wikipedia data sets. An ensemble of two FS-RNNs achieves better BPC performance than the best known compression algorithm. Further experiments provided evidence that the Slow cell enables the network to learn long-term dependencies, while the Fast cells enable the network to quickly adapt to unexpected inputs and learn complex transition functions from one time step to the next.

Our FS-RNN architecture provides a general framework for connecting RNN cells as any type of RNN cell can be used as building block. Thus, there is a lot of flexibility in applying the architecture to different tasks. For instance using RNN cells with good long-term memory, like EURNNs [21] or NARX RNNs [26, 8], for the Slow cell might boost the long-term memory of the FS-RNN

architecture. Therefore, the FS-RNN architecture might improve performance in many different applications.

**Acknowledgments**

We thank Julian Zilly for many helpful discussions.

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
