[Reviews · NeurIPS 2017]

Reviewer 1



The paper proposed a new RNN structure called Fast-slow RNN and showed improved performance on a few language modeling data set. Strength: 1. The algorithm combines the advantages of a deeper transition matrix (fast RNN) and a shorter gradient path (slow RNN). 2. The algorithm is straightforward and can be applied to any RNN cells. Weakness: 1. I find the first two sections of the paper hard to read. The author stacked a number of previous approaches but failed to explain each method clearly. Here are some examples: (1) In line 43, I do not understand why the stacked LSTM in Fig 2(a) is "trivial" to convert to the sequential LSTM Fig2(b). Where are the h_{t-1}^{1..5} in Fig2(b)? What is h_{t-1} in Figure2(b)? (2) In line 96, I do not understand the sentence "our lower hierarchical layers zoom in time" and the sentence following that. 2. It seems to me that the multi-scale statement is a bit misleading, because the slow and fast RNN do not operate on different physical time scale, but rather on the logical time scale when the stacks are sequentialized in the graph. Therefore, the only benefit here seems to be the reduce of gradient path by the slow RNN. 3. To reduce the gradient path on stacked RNN, a simpler approach is to use the Residual Units or simply fully connect the stacked cells. However, there is no comparison or mention in the paper. 4. The experimental results do not contain standard deviations and therefore it is hard to judge the significance of the results.

Reviewer 2



This paper presents an RNN architecture that combines the advantage of stacked multiscale RNN for storing long-term dependencies with deep transition RNN for complex dynamics that allow quick adaptation to changes in the inputs. The architecture consists of typically four fast RNN cells (the paper uses LSTMs) for the lower deep transition layer and of one slow RNN upper cell that receives from faster cell 1 and updates the state of faster cell 2. The model is evaluated on PTB and enwiki8, where it achieves the lowest character-based perplexity when compared to similar-sized (#parameters or number of cells) architectures. The analysis of vanishing gradients and of cell change is insightful. One small note: should fig 1 say h_{t-1}^{F_k} ?

Reviewer 3



The paper presents a novel architecture Fast Slow Recurrent Neural Network (FS-RNN), which attempts to incorporates strengths of both multiscale RNNs and deep transition RNNs. The authors performed extensive empirical comparisons to different state-of-the-art RNN architectures. The authors also provided an open source implementation with publicly available datasets, which makes the mentioned experiments reproducible. It would be better if the authors compared their approach with other recent state-of-the-art systems like Tree-LSTM that captures hierarchical long term dependencies.